# Strengthening Breast Cancer Screening Mammography Services in Pakistan Using Islamabad Capital Territory as a Pilot Public Health Intervention

**DOI:** 10.3390/healthcare10061106

**Published:** 2022-06-14

**Authors:** Ayesha Isani Majeed, Assad Hafeez, Shahzad Ali Khan

**Affiliations:** 1Health Services Academy (HSA), Quaid-i-Azam University, Islamabad 44000, Pakistan; az10@hotmail.com (A.H.); shahzad@hsa.edu.pk (S.A.K.); 2Department of Radiology, Pakistan Institute of Medical Sciences (PIMS), Islamabad 44000, Pakistan

**Keywords:** breast cancer, screening mammography, health system strengthening, awareness campaigns, cancer prevention

## Abstract

Late diagnosis of treatable breast cancer is the reason for higher breast cancer mortality. Until now, no public breast cancer facility has been established in the Islamabad Capital Territory. First, a Federal Breast Screening Center (FBSC) was established. Afterward, awareness campaigns about breast cancer were organized among the public. Subsequently, women above 40 years of age were provided with mammography screenings. Data were analyzed in SPSS version 22.0. An intervention was performed using a six tier approach to strengthening the health system. Utilizing the offices of the FBSC and the national breast cancer screening campaign, breast cancer awareness has become a national cause and is being advocated by the highest offices of the country. The number of females undergoing mammography has increased each year, starting from 39 in 2015 to 1403 in 2019. Most of the cases were BI-RAD I (*n* = 2201, 50.74%) followed by BI-RAD II (*n* = 864, 19.92%), BI-RAD III (*n* = 516, 11.89%), BI-RAD IV (*n* = 384, 8.85%), BI-RAD V (*n* = 161, 3.71%), and BI-RVAD VI (*n* = 60, 1.38%). The current study has theoretical and practical implications for the contemplation of policymakers. The FBSC can serve as a model center for the establishment of centers in other parts of the country, thereby promoting nationwide screening coverage.

## 1. Introduction

The health system comprises multiple entities including organizations, resources, institutions, legislation as well as people who are striving to improve health. Health system strengthening (HSS) refers to addressing and ameliorating the issues faced by the people, resources, organizations, and other components of the health system. The World Health Organization (WHO) has put forward a breakdown for health system strengthening (HSS) by dividing it into six components: health service delivery, workforce, health information system, medical technology, health financing, and leadership/governance. When assessing the situation of health care service delivery in Pakistan, the picture is not very promising. According to a Lancet study, Pakistan’s health system ranked number 154 while that of India is 145 out of 195 countries based on overall performance [1]. Therefore, it is necessary to make amendments to Pakistan’s health care system for improvement. Pakistan has a per capita health expenditure of around USD 18. Estimates indicate that approximately 25% of the deaths in Pakistan are attributed to non-communicable diseases (NCD). NCDs exhort a significant burden on individuals and the country’s economy. The National Action Plan for Prevention and Control of Non-Communicable Diseases (NAP-NCD) aims to control and prevent NCDs, thereby decreasing the mortality rate and economic burden on the country. Among all of NCDs, one of the biggest challenges is the rising mortality and incidence of breast cancer [2].

Breast cancer is a basic public health problem responsible for a continual increase in cancer-related morbidities around the globe. In both developed and developing countries, the prevalence and mortality rate of breast cancer is on the rise, majorly affecting women above the age of 40 [3]. According to GLOBOCON 2020, worldwide, breast cancer incidence has surpassed lung cancer with 2.3 million new reported cases in 2020. A total of 684,996 deaths were reported to be due to breast cancer all over the world, thus breast cancer is responsible for the highest death toll among all cancers in females [4]. Breast cancer mortality is greatest in 156 out of 185 countries. Comparatively, breast cancer death rates are greater in underdeveloped countries [5].

The Breast Health Global Initiative (BHGI) developed a paradigm for early breast cancer diagnosis and treatment in developing nations [6]. They studied how early detection and multimodal treatment affected breast cancer outcomes. Studies have indicated that early breast cancer diagnosis lowered cancer mortality. Mammography screening detects malignancies. The American College of Radiology (ACR) advises starting mammography at the age of 40 [7]. More frequent screenings increase the likelihood of detecting malignancies early and successfully treating them. Thus, mammography screening programs and resources are needed for early diagnosis [8].

Mostly, the late diagnosis of treatable breast cancer is the reason for the higher breast cancer mortality rate [9]. In Pakistan, there is a literature gap as well as the lack of a national cancer registry, resulting in limited information on breast cancer prevalence in different regions [10]. However, according to some studies, almost one in nine women in Pakistan suffers from breast cancer [3].

Therefore, in Pakistan, there is a dire need for a breast cancer screening program. Until 2015, no public breast cancer facility had been established in the country’s capital (i.e., Islamabad Capital Territory (ICT)). Some diagnostic machines were previously available in Nuclear Medicine, Oncology and Radiotherapy Institute (NORI) as well as some private breast cancer diagnostic facilities that were available to the public in the ICT. The Pakistan Institute of Medical Sciences (PIMS), the primary public health care facility in the ICT had one nonfunctional machine since 2005. This scenario presents a dire need for the establishment of an efficient yet cost-effective system for breast cancer screening in the Islamabad Capital Territory.

Apart from the establishment of a breast cancer screening facility, there is also a need for public awareness regarding the disease’s signs and symptoms, so that the people can perform breast self-examination (BSE) and clinical breast examination (CBE) for the early detection of breast cancer. Awareness relating to the need for regular mammography is also a prerequisite for early breast cancer diagnosis [11].

Factors responsible for breast cancer development must be identified and studied in detail as the knowledge of these factors can play a pivotal role in the early detection of breast cancer. Knowledge about these factors must also be made part of the awareness campaigns relating to breast cancer detection, as empowering women with knowledge about the causative factors in conjunction with breast cancer awareness can substantially increase a patient’s skill to carry out self-assessment for the early detection of breast cancer [12]. These practices will in turn benefit both the patient and the medical practitioners through an early presentation of the disease.

The study was conducted to strengthen the mammography screening system of Pakistan by using ICT as a pilot. Different aspects of the study include the analysis of the current situation of the health system of the ICT, setting up a free mammography service, awareness campaigns about breast cancer, the training of female health workers, and the analysis of this intervention.

## 2. Materials and Methods

The study was divided into different phases (Table 1). 

### 2.1. Development of Screening Centre

In the first phase, a budget of PKR 225 million was generated by the government to carry out breast cancer awareness activities in the ICT and establish a dedicated breast cancer screening center at the PIMS, bearing in mind the need for mammography screening to reduce the morbidity and mortality of breast cancer in the ICT. A 2D mammography machine of Metrionics, a 3D digital computed tomomammography with stereotactic biopsy machine from Hologic, and an ultrasound with Elastography software from GE healthcare was purchased.

### 2.2. Community Based Interventions

In the next phase, awareness campaigns about breast cancer were organized among the general public. In 2016 and 2017, multiple field visits in urban and rural Islamabad were carried out to spread awareness regarding breast cancer and its detection.

To create awareness in the rural population, a team of doctors and nurses were given training at the Pakistan Institute of Medical Sciences to recognize the signs and symptoms of breast cancer and to enable them to perform a proper clinical examination in the field. Lady health workers (LHWs) and lady health visitors (LHVs) recruited at the basic health units (BHUs) and rural health centers (RHCs) in the ICT were trained for these campaigns by general surgeons, radiologists, and gynecologists. The training included the awareness about the signs and symptoms of breast cancer, examination, management, and public awareness campaigns. LHWs and LHVs motivated the female population of selected areas to visit their closest BHUs and RHCs to attend lectures detailing the risk factors, warning signs, screening, self-examination, and therapeutic approaches to breast cancer. 

A referral system was introduced among the BHUs, RHCs, and FBSC, Islamabad. The experts also visited girls’ schools, colleges, and universities in the Islamabad Capital Territory. Among this team of experts, radiologists, general surgeons, genetic counselors, and gynecologists delivered presentations on all aspects of breast cancer. These lectures also consisted of the risk factors, signs and symptoms, self-examination, screening, and the role of early screening in the effective management of breast cancer. After attending the awareness seminar, women from the rural background of all ages were referred to PIMS Screening Center.

In the years 2018 and 2019, the focus from field visits was shifted to the utilization of press/print and electronic media. On multiple occasions between 2015 and 2019, walks were conducted for the awareness of breast cancer and the benefits of screening. 

### 2.3. Impact of Screening Mammography

The FBSC started operating in October 2015. It offers state-of-the-art free mammography services for all women over 40 years of age, and ultrasound facilities for women younger than 40 years of age. 

Subsequently, all women from the selected areas above 30 years of age were provided with general breast examination. All women above 40 were willing to undergo mammography screening were provided with free screening services. Women between 30 and 40 years of age and with some symptoms or lump on examination were also offered a mammography. The 2D-mammography screening was performed by a female technologist via the digital enhanced stereotactic mammography machine with biopsy facilities from Hologic. Two views (i.e., cranio-caudal (CC) and medio-lateral oblique (MLO)) were taken. Images were categorized according to the Breast Imaging Reporting and Data System (BIRADS) classification; fifth edition by post-graduate trainees and then verified by a consultant radiologist with a minimum of five years of experience [13]. The demographic data of these women were also collected. After three to six months, the BIRADS III lesions were followed with ultrasound. Patients with a family history were followed after three months while the rest were followed after six months as per the ACR guidelines. Fine needle aspiration cytology (FNAC) was offered to those women with high suspicion. Trucut biopsies were conducted for the BIRADS IV and V patients, and the specimens were sent for histopathological classification. Patients confirmed via histopathological classification were categorized as BIRADS VI. From 2018, women under the age of 40 were provided with sonomammography services. Ultrasound examination was performed by a sonologist with five years of experience. The BIRADs classification was also performed on sonomammograms.

The flowsheet diagram showing the operation of breast cancer screening center is shown in Figure 1 below.

Detailed data analysis was performed after five years of functioning of FBSC. Since 2D-mammography screening was started in 2015 while sonomammography in 2018, therefore, the data included the five year data of 2D-mammography (2015–2019) and 2 year data of sonomammography (2018–2019). Data were managed and analyzed in SPSS version 22.0. A year-wise record of screening mammography was analyzed. Cross-tabulation was performed to check the distribution of the BIRADs year-wise and age-wise.

## 3. Results

### 3.1. Health System Strengthening by Screening Center Development

The establishment of the breast cancer screening center at PIMS took place bearing in mind the six tiers of health system strengthening (HSS). The establishment of the center itself comprised the ‘health service delivery’ component of HSS. ‘Medical technology’ consisting of ultrasound with elastography software, 2D mammography machine, 3D digital computed tomomammography with stereotactic biopsy machines worth PKR 75 million were installed. All machines were integrated with a hospital management information system (HMIS). The woman could be traced via the PCN-number allotted once they reported to the hospital. ‘Health financing’ was available in the form of an approved budget of PKR 225 million from the Government. A complete ‘workforce’ for the efficient working of the center was developed according to Table 2. ‘Leadership and governance’ was provided via involving the office of the District Health Officer (DHO) and principals of educational institutes for their kind permission and support regarding the project.

### 3.2. Community Based Interventions

A total of 269 LHWs were trained to carry out clinical breast examinations. A total of 1670 people of different age groups underwent breast examinations as part of the awareness and screening sessions. A referral system was created to send women to FBCSC in need of further evaluation by screening modalities such as ultrasound and mammography. Table 3 and Table 4 show the record of such activities in a rural population in 2016 and 2017.

A total of 1000–1500 students of different age groups underwent these awareness sessions in urban Islamabad. In addition to this, a student volunteer program was also launched. Those willing to participate were made part of teams visiting the BHUs and RHCs. Apart from lady health workers (LHW), 10 lady health supervisors (LHS) were also trained. Apart from the LHW, 10 LHS and two ALHS were also trained. Table 5 shows the list of LHS trained for the ICT population:

The colleges and universities visited in the urban ICT are shown in Table 6:

Pamphlets and brochures were disseminated to spread awareness (Appendix A). Multiple journal clubs were arranged to share the new research and approaches in the field of breast cancer screening with the consultants and residents. 

A walk was arranged to raise awareness regarding breast cancer in October 2019. The federal minister for the Ministry of National Health Services Regulation and Coordination (MoNHSRC) participated and led the walk (Appendix A). A workshop was conducted from the platform of the Radiological Society of Pakistan (RSP). The concept of one-stop breast cancer was floated through this workshop. The surgeons and pathologists were also incorporated to ensure a multidisciplinary approach. The consultants and residents of surgery and pathology participated in this discussion and the mechanism of working a one-stop breast clinic was discussed in the workshop.

The primary researcher was appointed as the focal person for breast cancer awareness. A strong voice was raised from the presidency by the First Lady for breast cancer awareness. 

Letters were sent to federal ministers for their participation in breast cancer awareness campaigns. In the same year, the metro buses and stations were branded with breast cancer awareness messages. The office of the district health officer (DHO) was taken on board to formulate an effective plan for the training of LHWs. Utilizing the impact of cellular services on the general population, messages for breast cancer awareness were transmitted by the Pakistan Telecommunication Authority (PTA). The impact of media cannot be denied and it is considered as the fourth pillar of government. The awareness messages were broadcast over different TV channels of the benefits of breast cancer screening. Different talk shows and programs were also conducted.

Figure 2 shows the details of the number of patients visiting the center for mammography screening purposes:

As the results depict, we can see that the number of women undergoing mammograms increased every year. Month-wise analyses of all years were also performed, which are depicted in Figure 3:

The patients attending the scanning center were at the maximum in November, followed by December, April, March, and October. 

The possible reason for this would be the pleasant weather in March and April after the winter spell and the breast cancer awareness campaign in October, which led to an increase in the number of patients visiting the scanning center up to December before the winter set in. 

The total women imaged from October 2015 to December 2019 was 4337. The mean age for the women coming for mammography screening was 49.04 ± 8.93. Figure 4 shows the graphical representation of the age-wise distribution of women visiting for mammography screening.

The aged group most commonly attending the center was noted to be between 41 and 45 with the number being 1222 out of 4337, which was 28.17%, followed by the 46–50-year old age group with 1016 out of 4337, which was 23.42%.

Figure 5 shows the BIRADS categorization of women having mammogram screening.

Most of the cases were BIRAD I, which means normal without any findings. These cases accounted for 2201 out of 4337, which was 50.74%. The second most common category was BIRAD II, which was 864 out of 4337 (19.92%). BIRAD III was the third most common with 516 out of 4337, which made it 11.89%. The BIRAD IV (probably malignant) category accounted for 384 out of 4337 or 8.85%. The BIRAD V (definitely category) numbered 161 out of 4337 (3.71%). The BIRAD VI cases were 60 out of 4337, which was 1.38%.

The age group-wise distribution of the BIRADs is shown in Table 7.

An ultrasound machine was installed in 2017 at FBSC and became functional in 2018. A total of 3492 women underwent sonomammography during 2018 and 2019. The mean age of the women undergoing sonomammography was 31.83 ± 14 years. An increase in the trend was noted in the months of October and November. Figure 6 shows the detailed analysis of this trend.

The BIRADs categorization on sonommamography showed BIRADs I to be the most common type, followed by BIRADs II and III. BIRADs VI was found in 0.51% of the females. Figure 7 shows the detailed BIRADs categorization upon sonomammography.

## 4. Discussion

According to the World Health Organization (WHO), the health of an individual can be defined as “a state of complete physical, mental and social well-being and not merely the absence of disease or infirmity’’ [14]. In order to ensure adequate health at a population level, a health system is required that aims at ameliorating the average health of a population. “Health system” is defined by the WHO as all the organizations, institutions, resources, and people whose primary purpose is to improve their health [15]. To strengthen health systems means to address the key constraints related to health worker staffing, infrastructure, health commodities (such as equipment and medicines), logistics, tracking progress, and effective financing at all levels of the system functioning. The WHO laid out a framework for action on strengthening health systems with six building blocks: service delivery; health workforce; information; medical products, vaccines, and technologies; financing; and leadership and governance [14]. Moreover, in developing health care systems, new situations or public health crises often arise, leading toward a need for an ‘intervention tool’, which serves to strengthen the health care system for a particular purpose.

Pakistan is a low- to middle-income (LMIC) country. Its health care system is far from being able to sustain its gigantic population. Out of all the women’s health concerns in middle income countries, breast cancer has emerged as one of the biggest dangers to public health [16]. As for the rest of the world, breast cancer is one the leading causes of NCD related deaths in the female population of Pakistan [17] According to recent GLOBOCAN cancer statistics, breast cancer forms 24.5% of the total cancers. The ACR recommends mammography screening to be started from forty years to the control and timely management of the disease [18]. In well-developed countries, where 90% of the population has screening coverage, disease is usually diagnosed at stage I. However, in middle- and low-income countries with poor screening coverage, disease is diagnosed at late stages, thus results in poor prognosis. Delays in diagnosis are usually due to poor facilities, poverty, cultural and religious beliefs, and misconceptions [19].

Due to the high burden of breast cancer in Pakistan, this study was designed to help strengthen the health system of the ICT. The total female population of the ICT was 950,721 with 474,887 females residing in urban Islamabad and 475,840 females in rural areas. Until the start of this study, no public health care facility related to screening breast cancer had been established in the ICT, though private hospitals have some equipment that perform mammography upon payment. This gap justified the need for the establishment of an efficient public breast cancer screening facility in the ICT. For this purpose, breast cancer screening was used as an interventional tool in this study. 

In the first phase of the study, the FBSC was developed bearing in mind the six tiers of health system strengthening by the WHO. 

In the next phase, awareness to the general public about the establishment of such a system and its utility is required. This was conducted via community-based medical education (CBME), which is the delivery of medical education in a social context. It has been proved to be successful in instigating the importance of medical issues in the minds of the public in many studies. This health advocacy has a pertinent role in HSS [20]. Important early detection steps in this context include CBME, clinical breast examination, and resource adapted mammography screening [8]. Thus, bearing these factors in mind, the urban and rural populations in the ICT were targeted for awareness campaigns. Effective awareness campaigns consist of many components such as in-person visits by experts, doctors, etc. to the community as well as the usage of other communication channels such as media, text messaging, etc., which are all multiple parts of an effective awareness campaign. The use of these multiple components is pivotal to the success of a CBME campaign. 

In this study, multi-modal communication was used in the awareness campaign about the importance of early diagnosis and breast cancer screening utility for increased survival rates and the health of the patients. The purpose of this awareness campaign was to increase the number of patients visiting the FBSC so that this develops into an effective intervention for health system strengthening in the ICT.

The FBSC started operating in 2015 and since then, there has been an increased number of patients visiting the center and undergoing screening and mammography each year. Such an increasing trend in patients visiting the center is an effective translation of a successful awareness campaign. Globally, October is celebrated as breast cancer awareness month. The highest number of screening mammographies are conducted this month according to various statistics as a result of the increased community-based interventions this month. A study conducted on the attendance of women undergoing mammograms month-wise reported the highest number of mammograms in the month of October [21]. Our study results were in line with this observation, with the highest number of mammographies in November and December. After October, May and June were the other months with the highest attendance. They attributed this to pleasant weather conditions in their country. In our results, second peak attendance was noted in the months of March and April with the lowest attendance in June and July. This can be attributed to the pleasant weather of Islamabad in the months of March and April compared to the hot weather in June and July.

The main goal behind any screening program is the early diagnosis of breast cancer. In Asia, there is usually a late diagnosis and subsequently a high mortality rate for breast cancer. Only in some countries such as China, Korea, Taiwan, and Japan, are there population-based screening programs. The rest of the Asian countries usually have opportunistic screening programs. In Malaysia, a private setup showed a cancer diagnosis rate of only 0.39% [22]. In our study, histopathology-proven malignancy was detected in only 1.38% of the patients. This figure is comparable to other low-income countries that have inadequate screening coverage. A similar study in Nigeria reported a figure of 2% of histopathology proven malignancy among the screening mammography cases [23].

In contrast, countries with full coverage have reported high detection rates in contrast to countries without coverage. In the United States, 13% of cancer detection rates were reported during the results of the 2019–2020 screening (i.e., one in eight women has a risk of breast cancer) [24]. A good prognosis was evident by the fact that the majority of the patients had localized disease. A similar trend was observed in Europe where countries with 75–85% screening coverage such as Sweden, Finland, Denmark, and the UK have higher survival rates compared to countries with a screening coverage less than 50% (i.e., Serbia, Croatia, and Hungary) [25].

## 5. Conclusions

The current study has theoretical and practical implications for the contemplation of policymakers and experts. The Federal Breast Screening Center can serve as a model center for the establishment of centers in other parts of the country, thereby promoting nationwide screening coverage. Additionally, there is a need to focus on the continuous community-based medical education of women regarding breast cancer screening to prevent its lethal outcomes. For the prevention of disease at the national level, there is a need to establish a national-level free screening program. Only this can help in early diagnosis, thereby reducing the morbidity and mortality related to breast cancer in Pakistan. 

## Figures and Tables

**Figure 1 healthcare-10-01106-f001:**
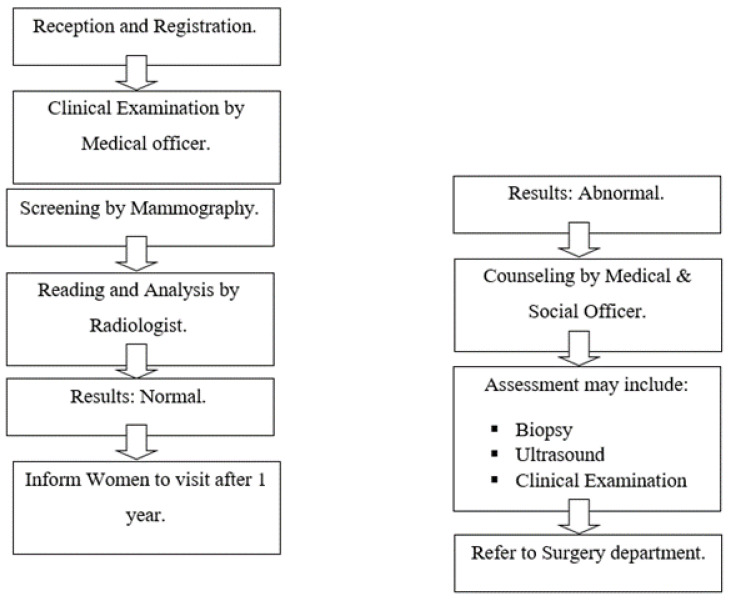
Operation of Breast Cancer Screening Center.

**Figure 2 healthcare-10-01106-f002:**
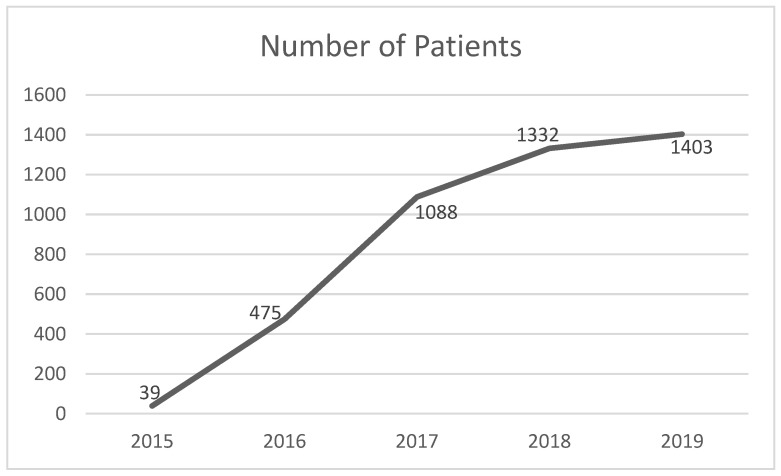
The year-wise number of patients visiting the center for mammography screening purposes.

**Figure 3 healthcare-10-01106-f003:**
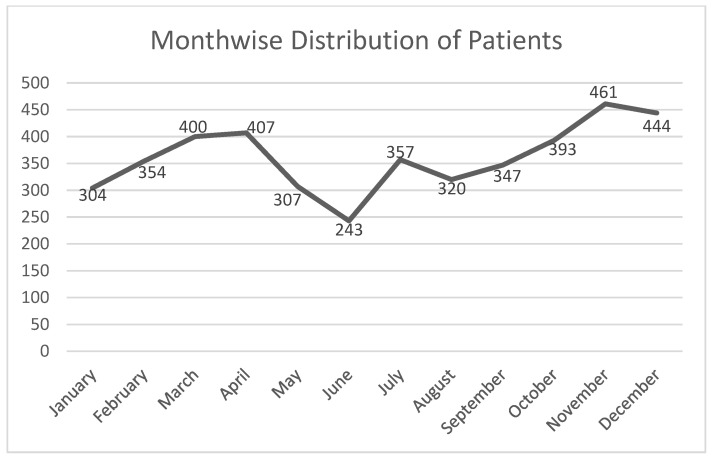
The month-wise analyses of the number of women visiting the screening center.

**Figure 4 healthcare-10-01106-f004:**
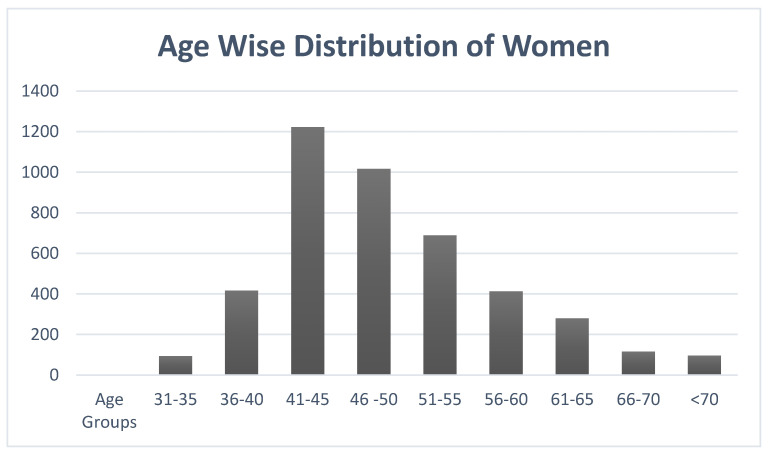
The age-wise distribution of women visiting for mammography screening.

**Figure 5 healthcare-10-01106-f005:**
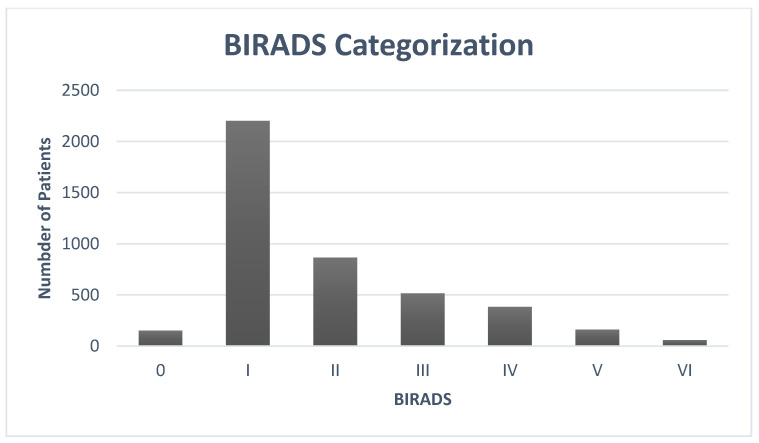
The BIRADS categorization of women with mammogram screening.

**Figure 6 healthcare-10-01106-f006:**
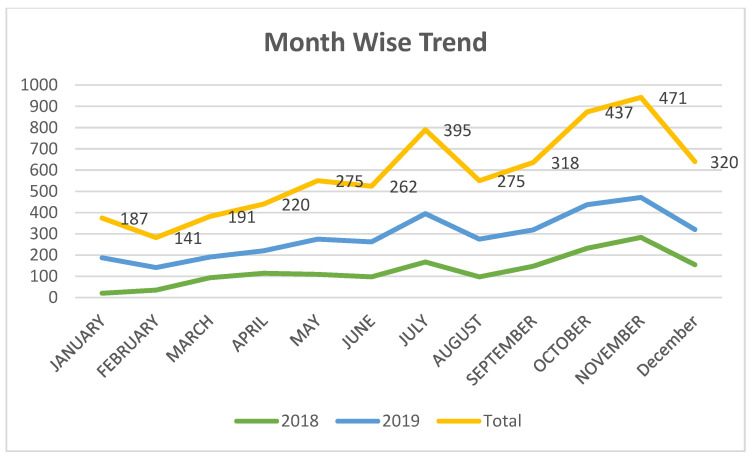
The year versus month-wise trend of women visiting for sonomammography.

**Figure 7 healthcare-10-01106-f007:**
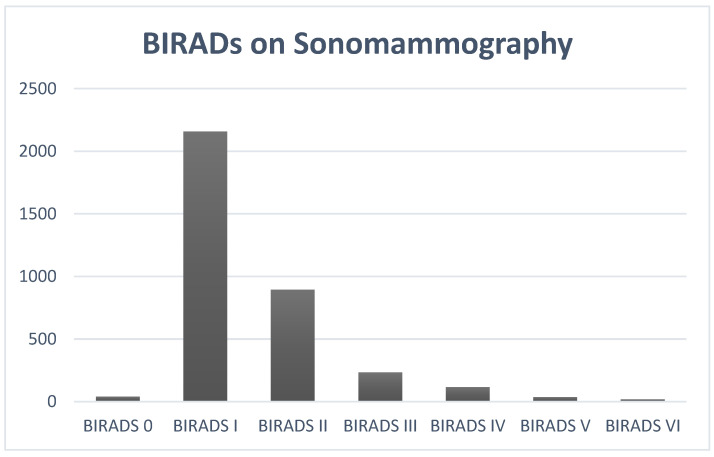
The BIRAD categorization on sonomammography.

**Table 1 healthcare-10-01106-t001:** The study design depicting the phase wise distribution.

Phase	Objective	Methodology
Phase 1 (2013–15)	Breast Cancer Screening Interventions Using 6 Tiers of Health System Strengthening in ICT	A well-equipped center for the diagnosis of breast cancer was developed bearing in mind the six basic tiers of health system strengthening. The center was equipped with two mammography screening facilities and one ultrasound machine. The center became functional, and a referral system was created.
Phase 2 (2016–2018)	Community Based Medical Education.	To bring awareness among the female population of the capital territory, rural areas, and female educational institutions including higher schools, colleges, and universities were targeted. Female health workers were also trained.
Phase 3 (2015–2020)	Impact of Screening Mammography	Mammography screening started and the records were maintained and analyzed after five years of performance.

**Table 2 healthcare-10-01106-t002:** The expenditure utilization breakdown in different components.

Expenditure Breakdown (in Million PKR)	
Civil Works	58.32
Machinery Equipment/Furniture	87.86
Establishment	7.950
Project Management Unit	17.89
Health Education	34.45
General	1.50
Transport	4.53
POL	3.00
Computer and Equipment	4.00
Purchase of Furniture and Fixture	3.00
Repair and Maintenance	1.00
Utilities	1.00
Other	0.50
Total	225.00

**Table 3 healthcare-10-01106-t003:** The activities carried out in rural Islamabad in 2016.

Location	Visit	Accompanying Healthcare Workers	Number of People Reached
Rawat BHU	January 2016	19	45
Sihala RHC	August 2016	16	90
Sihala RHC	August 2016	16	110
Shah Allah Ditta BHU	August 2016	12	40
Shah Allah Ditta BHU	September 2016	12	43
Chirrah, BHU	September 2016	16	47
Tarlai, RHC	October 2016	17	111
Pind Begwal, BHU	October 2016	14	48
Tumair, BHU	November 2016	12	43
Bhara Kahu, RHC	November 2016	16	97
Shah Allah Ditta BHU	November 2016	16	50
Rawat BHU	December 2016	19	45
BHU, Humak	December 2016	13	48
Shadrah, BHU	December 2016	14	35

**Table 4 healthcare-10-01106-t004:** The activities carried out in rural Islamabad in 2017.

Location	Visit	Accompanying Health Workers	No. of Patients Reached
Gokina, BHU	January 2017	12	41
Bhimber Trar, BHU	January 2017	13	45
Jagiot. BHU	February 2017	13	40
Gagri, BHU	February 2017	12	43
Sihala, RHC	October 2017	16	98
Shah Allah Ditta & Gagri, BHU	October 2017	16	42
Chirrah, BHU	October 2017	12	39
Tarlai, RHC	October 2017	17	110
Pind Begwal and Bhimber Tarar, BHU	October 2017	16	49
Tumair, BHU	October 2017	11	47
Bhara Kahu, RHC	October 2017	13	120
Rawat, BHU	October 2017	12	46
Humak, BHU	October 2017	12	50
Shahdrah, BHU	October 2017	14	48

**Table 5 healthcare-10-01106-t005:** The LHS trained for the rural sector.

Supervisors	AREA
LHS/ADC	MC G-9 + CHC SAD + All Centers
LHS	BHU Charra + BHU Tumair
LHS	BHU Jageyot
LHS	BHU Pind Bagwal + BHU Phulgran
LHS	BHU Bhukkar + BHU Bimber Tarar
LHS	RHC Sihala + BHU Gajri
LHS	BHU Rawat
LHS	BHU Sohan
LHS	MC G-7 + Gokina Dispensary
LHS	RHC Bharakhu + BHU Shahdara
ALHS	RHC Tarlai
ALHS	BHU Jhang Sayyadian + BHU Kirpa

**Table 6 healthcare-10-01106-t006:** The institutes visited by the doctors’ team to create awareness in the urban population of Islamabad.

Sr. No.	Institute	No. of Participants.
I	Islamabad Model College for Girls, F.6/2, Islamabad	88
II	Federal Government College for Women, F.7/2, Islamabad (95)	95
III	Islamabad Model College for Girls, F.7/4, Islamabad (107)	107
IV	Islamabad Model College for Girls, G.8/4 (T&T colony), Islamabad	110
V	Islamabad Model College for Girls, I-8/4, Islamabad	86
VI	Federal College For Women G.10/4 Islamabad	95
VII	Federal College For Women F.7/4 (Margalla) Islamabad	90
VIII	Islamabad College for Girls F.6/2 Islamabad	82
IX	Beacon House H-8, Islamabad	103
X	International Islamic University, Islamabad	109
XI	Quaid-e- Azam University, Islamabad	150
XII	Bahria University, Islamabad	120
XIII	Comsats, Islamabad.	200

**Table 7 healthcare-10-01106-t007:** The age group-wise distribution of BIRADs.

Age Groups	BIRAD 0	BIRAD I	BIRAD II	BIRAD III	BIRAD IV	BIRAD V	BIRAD VI	Total
≥30	3	9	0	5	4	2	6	29
31–35	7	32	7	17	17	5	8	93
36–40	16	233	63	43	42	15	4	416
41–45	44	652	248	165	77	28	8	1222
46–50	37	518	216	121	87	24	13	1016
51–55	30	342	148	73	59	20	16	688
56–60	9	201	89	41	40	26	6	412
61–65	4	141	55	29	27	20	3	279
66–70	2	48	24	16	14	9	2	115
<70	2	28	20	11	19	14	2	96
Total	151	2195	870	516	382	161	62	4337

## Data Availability

Data will be provided upon reasonable request.

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
