# Peer review of "Strengthening Breast Cancer Screening Mammography Services in Pakistan Using Islamabad Capital Territory as a Pilot Public Health Intervention"

_healthcare, 2022, doi:10.3390/healthcare10061106_

Round 1

Reviewer 1 Report

This study reported a public health project of establishment of breast screening system in Pakistan, which brings new literature to the research topic. However the structure of this paper makes it hard to read because phase 1-2 are better presented in report format rather than original research article. Suggest to separate phases 1& 2 (Centre establishment and Public campaign) and phase 3; describing phases 1&2 in reporting format to introduce how the project/program was established and organised (many information in Discussion can be used), and reporting phase 3 where recruiting women for screening in research article format to analysis the data of screened women.

Title: do not use semicolon in a title.

Introduction:

  • Line28-32: ‘The Health system’ – why Health with a capital ‘H’? Is this refereeing to s specific system? If so, an explanation is needed here because readers are not familiar with it.
  • Similar issues are found in line 38-39, 45,
  • Line 46: what is NAP? No full name for this abbreviation.
  • Line 55-57, is the incidence a worldwide incidence or for a specific region?
  • Line 59-60: BHGI needs a reference.
  • Line 77: NORI? No full name for this abbreviation.

Materials and methods:

  • Author noted different phases for this study but there is no reporting of timeframe of each phase.
  • Line 128-135: lack of necessary details in this phase 3. For example, which women were provided with general breast examination and screening mammography? Women from the whole region in ICT or specific regions or with specific characterises (e.g. from school). Also, there is no information about who provided these examinations, radiologists or breast physicians or breast physicians or technicians, .etc. Besides, there is no information regarding the reporting process, e.g. who did the reporting, which version of BIRADS. In addition, lack of details on how each step happened.

Results:

  • Organisation of this section (on page 6) is not very well and hard to follow.
  • Line 169: which table?
  • Figure 2: which year data is this from?
  • Line 235-236 should go to Methods.
  • Figure 5: if authors would like to present this figure in B&W, please consider using different pattern of lines for the 3 years because I can’t figure out which one is which on the current presentation.

Discussion:

  • Line 319: needs references.
  • Line 340-342: the conclusion is too strong, and can not be directly concluded from this study only because in this study, the authors did not consider cost-effectiveness of the screening program.

Languages:

  • Many typos or style errors throughout the whole paper, including abstract, need to be updated.
  • Capital letters were noted for many words throughout the paper. There is no need to use capital letters for any word if it is refereeing to a specific subject. Please correct them.
  • Many abbreviations showed up without its full name/explanation, e.g. NORI, BIRADS, FNAC, POL, PKR, FBCSC, MoNHSRC, .etc.

Reviewer 2 Report

Dear Authors,

I appreciate the efforts to describe the scenario in your country. Your paper could be a stimulus towards an in-depth evaluation of the argument provided and to promote breast cancer screening in your country.

Nevertheless, I think that your paper could be accepted for publication in “Healthcare” after the following revisions:

11. The paper needs to be extensively English-checked. There are many typos alongside the paper and some sentences are unclear. For example, in the abstract: what does it mean “being a focal person”, line 16? Furthermore, please write “classified as BI-RADS 1, BI-RADS 2” etc. and correct the typing error in the abstract and along the paper (f. e. BIRAD, BIRVAD and so on).

22. Introduction: there are some redundancies in this section. Please, merge and rewrite some paragraphs by highlighting the importance of a well-defined screening program in your country as well as the need for a public awareness towards this disease due to the current limitations. For example, lines 28-34 and 59-71 are redundant and that concept about health system and early diagnosis could be expressed in a more synthetic way. Furthermore, please, clearly highlight also the major importance of digital mammography as gold standard for screening examination in women aged more than 40 years old while Breast Self-Examination and Clinical Breast Examinations could be considered as relevant only as a first step towards a more in-depth instrumental evaluation in that age-range (mainly for awareness purpose) but they could be considered as first level exam in women aged less than 40 years old in addiction to ultrasonography that is the gold standard in this age-range [https://www.cdc.gov/cancer/breast/pdf/breast-cancer-screening-guidelines-508.pdf].

33. Materials and Methods/Results: these are the most critical sections. I think that some paragraphs of the results section should be merged and moved to the “materials and methods” section as well as a better definition of the breast imaging techniques performed should be provided. Evaluate also the possibility to add sub-paragraphs.

44. Lines 102-106. The information about Ethical Board and Informed consent will be reported in the specific section at the end of the paper. So consider to delete them from here. Indeed, for example, this section could start with the sentence “This study was divided into four different phases” then lines 143-146 “A budget…ICT”.

55. Lines from 106 to 111 contain info about the first phase but more relevant data are described along the results section (lines 146 to 158). Please rearrange and merge these paragraphs to avoid repetition. Describe the methods in the specific section and add only the results (economical and practical consequences of the intervention) in the other one.

66. Lines 112 to 127 contain data about the second phase. Also in this case, some information are provided in the results section. Describe the methods in the specific section and add only the results (referred to table 1 and 2) and practical consequences in the other one.

77. Lines 128-135 contain info about the third phase. Please, could you better express the imaging modalities performed? F.e. What kind of mammography have you performed (2D digital, digital breast tomosynthesis) and in which projections (MLO, CC?). Have you performed a single or a double reading?

88. Please, clarify the acronyms BI-RADS and FNAC. Also add a note when you talk of BI-RADS classification (line 130).

99. Line 131. Why did you performed ultrasound after 3 months?

110. Lines 132-133. What kind of biopsies have you performed? Core Needle Biopsy, VABB?

111. Lines 134-135. Irrelevant if you add a note on line 130.

112. Clarify the pathway used for patients. Check for example figure 1 of this ePOS https://dx.doi.org/10.1594/ecr2017/C-0235  

113. Lines 136-137 should contain the info about the fourth phase. Please, explicit when you have evaluated the BI-RADS classification (for mammography and ultrasound).

114. Results: lines 141-142. I’m sorry but I don’t understand the reason why the sentence “No mammography…ICT” is written here. I think that it could be considered as a relevant aspect to be described in the introduction or in the discussion.

115. Results: as already suggested, in this section add only the results and practical consequences of the intervention (content of Tables and figures).

116. Line 151. Which currency?

117.   Line 169. Which table?

118.   Lines 217-218. ..which is depicted in figure 2. Avoid “in form of line graph”.

119. Lines 223-225. This sentence seems to be in contradiction with lines 310-316. Please, clarify in both sections.

220. Lines 235-236. Single or double-reading. As already suggested, move this info in the previous section.

221. Line 248. …shown in table 3. Avoid “in tabular form”.

222. Lines 252-254. Please, check the English language: “A total of…” What kind of patients are these ones? Second level examination after mammography or first level examinations?

223. Discussion: I think that this section should be expanded. Please avoid repetitions and focus on the relevance of your results according to criticism of your country. If you have data about the number and type of cancer detected (number of early breast cancers, locally advanced etc.), consider to add them. Lines 274-285 are redundant, consider to delete them.

Kind Regards

Round 2

Reviewer 2 Report

Dear Authors,

I congratulate you for the imporvements to your paper.
Please, note that on line 274, the number of the table is missing.
After the correction of this tiny point, I think that the paper could be accepted in current form.

Sincerely